# Dynamic risk assessment of hospital oxygen supply system by HAZOP and intuitionistic fuzzy

Yeganeh Yousofnejad[1], Fatemeh Afsari[2], Mahboubeh Es'haghi[3]*

1 Student Research Committee, Faculty of Public Health, Kerman University of Medical Sciences, Kerman, Iran, 2 Department of Computer Engineering, Faculty of Engineering, Shahid Bahonar University of Kerman, Kerman, Iran, 3 Department of Occupational Health Engineering and Safety at Work, Faculty of Public Health, Kerman University of Medical Sciences, Kerman, Iran

* esaghij@gmail.com

## Abstract

Events such as oxygen leakage in the oxygen generation systems can have severe consequences, such as fire and explosion. In addition, the disruption in the oxygenation systems can lead to a threat to patients' lives. Thus, this study aimed to identify the significant deviations in the oxygen supply system as critical equipment at hospitals based on the Hazard and Operability (HAZOP) method. Despite the advantages of risk assessment techniques, hazard identification techniques are still being utilized with deterministic and unreliable values and have a completely static nature. Therefore, using dynamic techniques to overcome intrinsic ambiguity in the risk assessment process through fuzzy sets has been recommended. Additionally, we proposed the HAZOP methodology to integrate with the intuitionistic fuzzy system for assessing the medical oxygen supply system using Pressure Swing Absorbance technology as a proactive approach. The results showed that the intuitionistic fuzzy approach, combined with the risk assessment method, is a suitable tool to eliminate uncertainty, improve decision-making, and result in more detailed and accurate findings. The approach adopted in this study can be used as a needs assessment tool to optimize maintenance programs and provide the necessary training for the staff, maintenance operators, and medical equipment managers.

## 1. Introduction

The National Fire Protection Association (NFPA: 2009 to 2013), on average, reported 5,750 fires each year in healthcare centers [1]. What makes the issue of safety in hospitals is that there are a considerable number of sick and disabled people. Thus, fires can seriously threaten patients' lives and interrupt medical treatment [2].

Hospital fires lead to high economic losses due to damage caused to medical equipment and hospital buildings [3]. One of the most critical pieces of equipment that have an essential role in the safety of patients is oxygen generators and ventilators in hospitals [4, 5]. In addition, the onset of the corona pandemic and high demands for oxygen consumption has already

**Data Availability Statement:** All relevant data are within the paper.

**Funding:** This research was supported by Kerman University of Medical Sciences by Ethical Code (ID: IR.KMU.REC.1399.500).

**Competing interests:** The authors have declared that no competing interests exist.

**Abbreviations:** HAZOP, Hazard and Operability Study; P&ID, Pipe and Instrument Diagram; PSA, pressure swing absorbance; TIFN, triangular intuitionistic fuzzy numbers.

increased the number of hospital concentrators [6]. Despite the medical benefits of this system, the increasing demand for oxygen consumption has been related to some incidents at medical centers. Looking back at the previous incidents that occurred worldwide, we can see that many underlying causes of fires were related to ventilators and oxygen generators. The study by Chowdhury (2014) [7] showed that high-concentration oxygen leakage from generators and ventilators was the primary source of fires and explosions in medical centers. Additionally, the explosion of oxygen cylinders in a hospital caused the injury of one patient in 2011. In addition, a ventilator and medical oxygen pipeline explosion damaged patients in India.

Furthermore, oxygen leakage in a central oxygen supply system caused an explosion in a hospital in South Africa [8]. A study conducted by Deleris et al. (2006) [9] in medical oxygen supply systems from the viewpoint of patient safety found that oxygen system failures led to more than forty-four fatalities of patients over thirty years. What is evident from these statements is that a comprehensive risk assessment of oxygen systems and identification of potential hazards is essential with an active approach.

One of the appropriate techniques to assess the risk of process equipment such as a hospital oxygen system is to use Hazard and Operability Study (HAZOP), which has the highest acceptability application in process industries [10]. As a qualitative technique, the HAZOP can review and identify potential deviations and operability problems using standardized guide-words and operational parameters in a system [11]. In this approach, the process equipment plant divides into smaller parts called nodes using a P&ID [12]. Then, the possible deviations in this process identify by integrating the operational parameters (such as temperature, pressure, flow, viscosity, PH, and so on) and guide words (none, more, less, reverse, as well as, and so on [13]. Willey et al. (2020) [13] showed that the HAZOP is practical for assessing and determining potential risks in process industries such as chemical plants. In addition, the study conducted by Oliveira and Ruppenthal (2018) [14] in hospital boilers using the HAZOP technique found that failures of components in the boiler system can lead to a high probability of fire and explosion occurrence.

Despite the advantages of risk assessment techniques, hazard identification techniques are still being utilized with deterministic and unreliable risk values and have a completely static nature. Thus, it has been recommended to use dynamic risk assessment techniques to overcome these limitations, which some researchers have accepted in recent years. For instance, Xin, Khan, and Ahmed (2017) [15] utilized a dynamic hazard assessment to identify critical risks and scenarios through the Bayesian network. Wei, Laibin, and Jinqiu (2009) [16] used the integration of HAZOP analysis and Fuzzy Information Fusion theory to overcome the challenges of quantitative information loss and make a decision under uncertain conditions in a gas compressor unit. This approach helped them achieve a more accurate and reliable safety system. Ilangkumaran and Thamizhselvan (2010) [17] also integrated HAZOP and Failure Mode Effect Analysis techniques using a fuzzy linguistics approach to identify and prioritize hazards in the petrochemical industry. This study showed that the dynamics of Risk prioritization could compensate for the weaknesses of the qualitative risk assessment approaches by fuzzy weight geometric mean. In addition, Cao et al. (2012) [18] performed a dynamic risk assessment by combining the HAZOP method and fuzzy logic in oil storage tanks that prioritized identified risks with unreliable data through fuzzy numbers. Hu et al. (2015) [19] also utilized the Dynamic Bayesian Network in the HAZOP method to overcome the uncertain data for risks in the petrochemical industry. The mentioned studies showed that the fuzzy set theory could be utilized in a wide range of fields in which the information for decisions is imprecise or incomplete. In addition to the mentioned studies above, some studies applied the new approach called intuitionistic fuzzy set for dynamic risk assessment, which provides a flexible and convenient solution by utilizing mathematical and statistical language in complex systems.

This approach aims to make results more accessible and understandable in which the input information and variables are subjective, qualitative, ambiguous, and uncertain [20]. One of the main strengths of this method is that it considers membership and non-membership functions of variables simultaneously [21]. Kabir et al. (2020) [20] integrated intuitionistic fuzzy set and fault tree analysis to analyze a fuel distribution system in a ship with uncertain data. This integrated technique showed that the proposed method provided more accurate output than the conventional method in a complex and dynamic system. Additionally, Viegas et al. (2020) [22] merged HAZOP and intuitionistic fuzzy sets for better decisions under the inherent uncertainty of the traditional HAZOP, which presented a more realistic perception of the levels of risk. Therefore, we considered the intrinsic ambiguity in the risk assessment process using the intuitionistic fuzzy approach.

According to the importance of the medical oxygen system in hospital environments, the present study was conducted in one hospital to perform a dynamic risk assessment by integrating the HAZOP technique with the intuitionistic fuzzy approach. This study consists of five sections. Section 2 presents the method and objective of the research and provides information on the medical oxygen system, HAZOP method, and intuitionistic fuzzy approach. Section 3 presents the results of the study. Section 4 discusses the findings and presents the implications and limitations of the study. Finally, section 5 offers the conclusion.

## 2. Material and methods

This study was performed in a hospital in Kerman province of Iran in 2020. The experts of this study consisted of occupational health and safety experts, artificial intelligence, medical equipment manager, hospital crisis manager, and device operators to perform the risk assessment. In this research, we assessed the risk assessment of an oxygen supply system with the PSA technology, which did not have human data or confidential information. Thus, the consent of the participants in the meetings was verbal.

### 2.1 The medical Oxygen concentrator system by pressure swing adsorption technology

Oxygen (O2), a colorless and odorless gas, occupies almost twenty-one percent of the atmosphere. Oxygen is a non-combustible gas, but it actively supports the burning of combustible materials. In addition, the combination of some chemical substances, such as oil and grease, with oxygen will cause fire and explosion accidents [23]. By considering the vital role of oxygen in the medical care system, it is necessary to have a comprehensive risk assessment of the oxygen-generating apparatus. The PSA process is a standard technology that supplies oxygen-enriched air under pressure through adsorption steps in medical centers. The medical oxygen concentrator operation by the PSA technology consists of an Oil-injected Rotary Screw Air Compressor, Refrigerated Compressed-air Dryer, air receiver, oxygen generators, an oxygen storage tank, and some valves and filters. It must be mentioned the oxygen concentrator apparatus under study can supply oxygen between 240 liters per minute with more than 90 percent purity [24].

**2.1.1 Description of the PSA system.** The air compressors are used for sucking and compressing volumes of the ambient air by oil injected-rotary screw compressors in hospital applications. The compressed air in the PSA system must be dry to avoid repair costs and equipment failure. Thus, compressed air dryers are mainly applied to remove the possible amount of moisture from the air stream. After that, compressed air is stored in a tank to ensure the PSA system's continuous and steady operation and provides a pressure-equalizing reservoir of compressed air, which is located before the oxygen generators. An oxygen generator

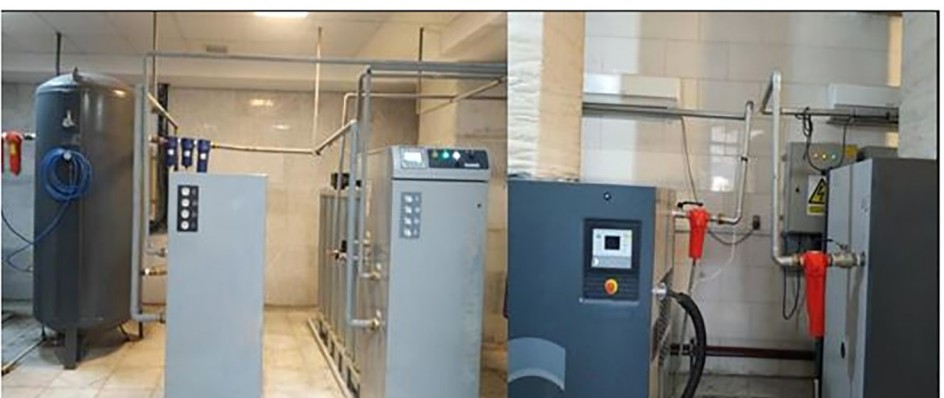

**Fig 1. The picture of the PSA technology in the hospital.**

has been utilized to supply purity oxygen-enriched gas by removing nitrogen gas from the compressed air through the adsorption separation. The enriched oxygen stream gets conserved in a storage tank with a pressure of about 7.5 bars and purity above 90%, which helps reach a steady enriched oxygen stream. It then flows out for consumption through the oxygen discharge pipeline [24, 25].

In addition to the previous explanation, there are two pressure and temperature sensors as electronic measurement devices in the PSA system. In this system, both sensors are used to measure the changes in the parameters. Then, the sensors' data are monitored and recorded in the logbook daily by the operators. In addition, a blade rotating fan at the end of the condenser is located, and which pressures switch controls that. The primary duty of the fan is to reduce the oil temperature and change the refrigerant phase from gas to liquid state to re-enter the air conditioner. Besides, a pressure regulator is utilized to adjust the pressure to the defined value, which reduces the oxygen pressure from eight to four bars for consumption. A one-way valve is also installed to prevent a gas return to supplement the pressure regulator function in this structure. In addition, nine manual valves are used to take the oxygen production system off in an emergency, replace filters, do repairs, and perform other activities. According to explanations, it can be said that the PSA system operates with a considerable number of apparatuses under some specific operation parameters. As a result, these unique features cause it to be considered one of the critical hospital equipment, especially its vital role in the safety of the patients (Fig 1).

## 2.2 HAZOP study

As a qualitative approach, the HAZOP is a systematic and structured technique that evaluates and identifies potential process hazards that may present risks to people, equipment, environment, and operation. The basic principle of the HAZOP is that a system is safe when all the process parameters are in their designed ranges. After preparing the Pipe and Instrument Diagram (P&ID), the overall diagram of the process is broken down into smaller sections called "nodes." In this study, the experts selected the nodes during brainstorming sessions. For each node in the diagram, the experts were asked to identify all potential deviations from the design intent by applying a logical combination of standardized keywords and process parameters in the system [26].

$$\text{Guideword} + \text{Parameter} = \text{Deviation}$$

**Table 1. Risk assessment matrix [28].**

| Risk Assessment Matrix | | Severity | | | | |
|---|---|---|---|---|---|---|
| | | Insignificant | Minor | Moderate | Major | Catastrophic |
| **Probability** | | 1 | 2 | 3 | 4 | 5 |
| Almost Certain | 5 | 5 | 10 | 15 | 20 | 25 |
| Likely | 4 | 4 | 8 | 12 | 16 | 20 |
| Possible | 3 | 3 | 6 | 9 | 12 | 15 |
| Unlikely | 2 | 2 | 4 | 6 | 8 | 10 |
| Rare | 1 | 1 | 2 | 3 | 4 | 5 |
| **Risk Level Ranking** | | | | | | |
| **Definition** | | | | | **Guidance** | Category |
| Low level- routine actions are required. | | | | | 1–3 | 1 |
| Medium level- specifies actions are required. | | | | | 4–6 | 2 |
| High level- senior actions are required. | | | | | 8–12 | 3 |
| Extreme level- immediate actions are required. | | | | | 15–25 | 4 |

$$over(keyword) + temperature(process\ parameter) = over\ temperature(deviation)$$

According to ISO 14971 (2019) [27], risk consists of two dimensions, including probability and severity (risk = probability × severity), in the risk assessment approach. Therefore, the experts were asked to determine the possibilities of incidents and severity due to events, consider the underlying causes of deviations, identify the existing safeguards, and propose additional precautions to reduce unacceptable risk levels to an acceptable level if necessary. Furthermore, the probability and severity of all deviations were estimated and ranked according to the standard risk matrix (Table 1) [28]. According to the 5×5 risk matrix, the five probabilities and severity categories are classified into four risk evaluation levels.

## 2.3 Intuitionistic fuzzy sets

The theory of fuzzy sets was introduced by Zadeh [29], which has been remarkably used in many different fields of study. The fuzzy set was presented to handle vagueness due to incomplete and uncertain information, which the classical sets could not consider. The concept of the fuzzy set is based on the membership functions defined for the linguistic variables that assign a value between zero and one as the membership degree of a variable. One of the drawbacks of the fuzzy sets is that there may be some hesitation about a variable's membership and non-membership degrees because a variable's non-membership degree may not be equal to one minus the membership degree. Thus, a generalization of the fuzzy set is introduced by Atanassov [30] called the intuitionistic fuzzy set. In this theory, a hesitation degree is defined by subtracting one from the sum of membership and non-membership degrees. Thus, it was more logical to utilize intuitionistic fuzzy linguistics to express hesitancy in the risk assessment process (Fig 2).

**2.3.1 Fuzzy systems.** A fuzzy system involves fuzzification, fuzzy inference using fuzzy if-then rules, and defuzzification steps. The fuzzification step includes the definition of the linguistic variables, which converts the crisp values to vague terms. In addition, the if-then rules make the relationship between the input and output variables. Then, the defuzzification process is utilized to convert the fuzzy values to crisp numbers [29].

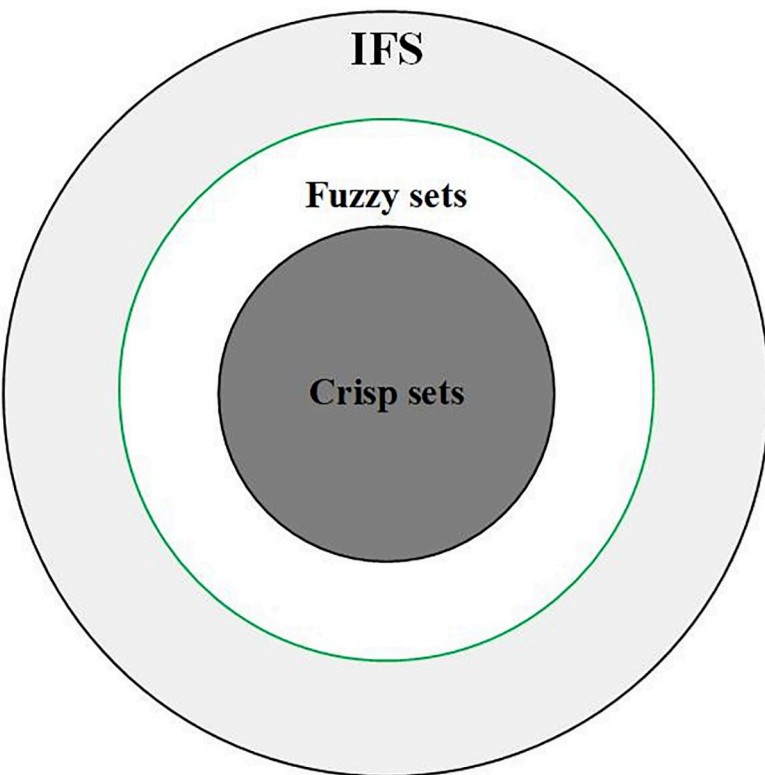

**Fig 2. Relationships between crisp, fuzzy, and intuitive fuzzy values.**

**2.3.2 Linguistic variable.**  In the fuzzy concept, each variable is defined using a quadruple including $\langle X, T, U, M \rangle$. Where, $X$ represents the linguistic variable $T$, the term set, is the set of linguistic values. Moreover, all fuzzy sets corresponding to the linguistic values of the variable $X$ are defined on the domain $U$. Finally, $M$ as a set of fuzzy rules relating input and output linguistic variables [31].

**2.3.3 Intuitionistic fuzzy sets.**  The intuitionistic fuzzy set is characterized by two different functions, including the membership and non-membership functions, which map any crisp value to values between zero and one. If we consider $U$ as the universe of discourse, then an intuitionistic fuzzy set $A$ is drawn from the $U$ is defined as follows, where the functions, $\mu_A$, $\nu_A \colon U \to [0,1]$, are the membership and non-membership functions, respectively [30].

$$A = \{(u, \mu_A(u), \nu_A(u)) \mid u \in U\}$$

Also, $0 \leq \mu_A(u) + \nu_A(u) \leq 1$ must be held for all $u \in U$ (30). In addition, the hesitation or indeterminacy degree of any $u \in U$ is defined as follow [30]:

$$\pi_A(u) = 1 - \mu_A(u) - \nu_A(u)$$

**2.3.4 Triangular intuitionistic fuzzy number.**  Following the work of Tsai et al.) 2015) [32], the present study applies the triangular intuitionistic fuzzy numbers (TIFN) as one of the most common types of intuitionistic fuzzy numbers to define the fuzzy values of the linguistic variables. This approach enables the conversion of conventional linguistic judgments into fuzzy scales to process the ambiguity of thought and decisions.

A TIFN $A = \langle(a, b, c); \alpha, \beta\rangle$, with the membership function $\mu_A$ and non-membership function $v_A$ is a special intuitionistic fuzzy set, in which the membership and non-membership functions are defined as follow [33]:

$$\mu_A = (u; a) = \begin{cases} 0 & u < a \\ \dfrac{a}{b-a}(u-a) & a \le u < b \\ \dfrac{a}{c-b}(c-u) & b \le u \le c \\ 0 & u > 0 \end{cases}$$

$$v_A = (u; \beta) = \begin{cases} 1 & u < a \\ \dfrac{1-\beta}{b-a}(b-u)+\beta & a \le u < b \\ \dfrac{1-\beta}{c-b}(c-u)+\beta & b \le u \le c \\ 1 & u > c \end{cases}$$

Where, $\alpha$ is the maximum value of the membership degree and $\beta$ is the minimum value of the non-membership degree, which are both values between zero and one such that $0 \le \alpha + \beta \le 1$. Fig 3 shows a triangular intuitionistic fuzzy number.

**2.3.5 Intuitionistic fuzzy inference system.** The intuitionistic fuzzy if-then rules can be used to model the opinion and knowledge of experts that are often expressed by linguistic words as an inference. Suppose $x = (x_1.x_2)$, and $y = (y_1.y_2)$ are two intuitionistic values where $x$, and $y$, are the membership values and $x$, and $y$, are the non-membership values. In the following, we explain several concepts necessary to construct an intuitionistic fuzzy inference system [31].

*2.3.5.1 Intuitionistic fuzzy triangular norms and co-norms.* The definitions of the t-norm $\mathcal{T}$ (triangular norm) and t-conorm $\mathcal{S}$ (triangular conorm) as intuitionistic fuzzy multiplication and summation operators are extended as follows [31]:

$$\mathcal{T}(x.y) = \big(T(x_1.x_2).S(y_1.y_2)\big)$$

$$\mathcal{S}(x.y) = \big(S(x_1.x_2).T(y_1.y_2)\big)$$

Where, $T(.,.)$ and $S(.,.)$ are the fuzzy t-norm and t-conorm, respectively.

*2.3.5.2 Intuitionistic minimum residual implication.* Let the t-norm be the minimum t-torm, $T(a, b) = \min\{a, b\}$. Hence, the intuitionistic minimum residual implication is written as follows [31]:

$$x_{\rightarrow \ell} y = \begin{cases} (1-y_2.y_2). & x_1 \le y_1.x_2 \le y_2 \\ (1.0). & x_1 \le y_1.x_2 > y_2 \\ ((y_2 \wedge (1-y_2).y_2 \vee (1-y_1)). & x_1 > y_1.x_2 \le y_2 \\ y_1(1-y_1). & x_1 > y_1.x_2 > y_2 \end{cases}$$

*2.3.5.3 Intuitionistic fuzzy if-then rules.* The intuitionistic fuzzy approach utilizes the intuitionistic fuzzy if-then rules by considering triplets including $A$, $B$, and $R$. In this triple composition, $A$ and $B$ are the antecedent (input variable) and the consequence (output variable), respectively. These two variables are connected together through a set of relations called $R$.

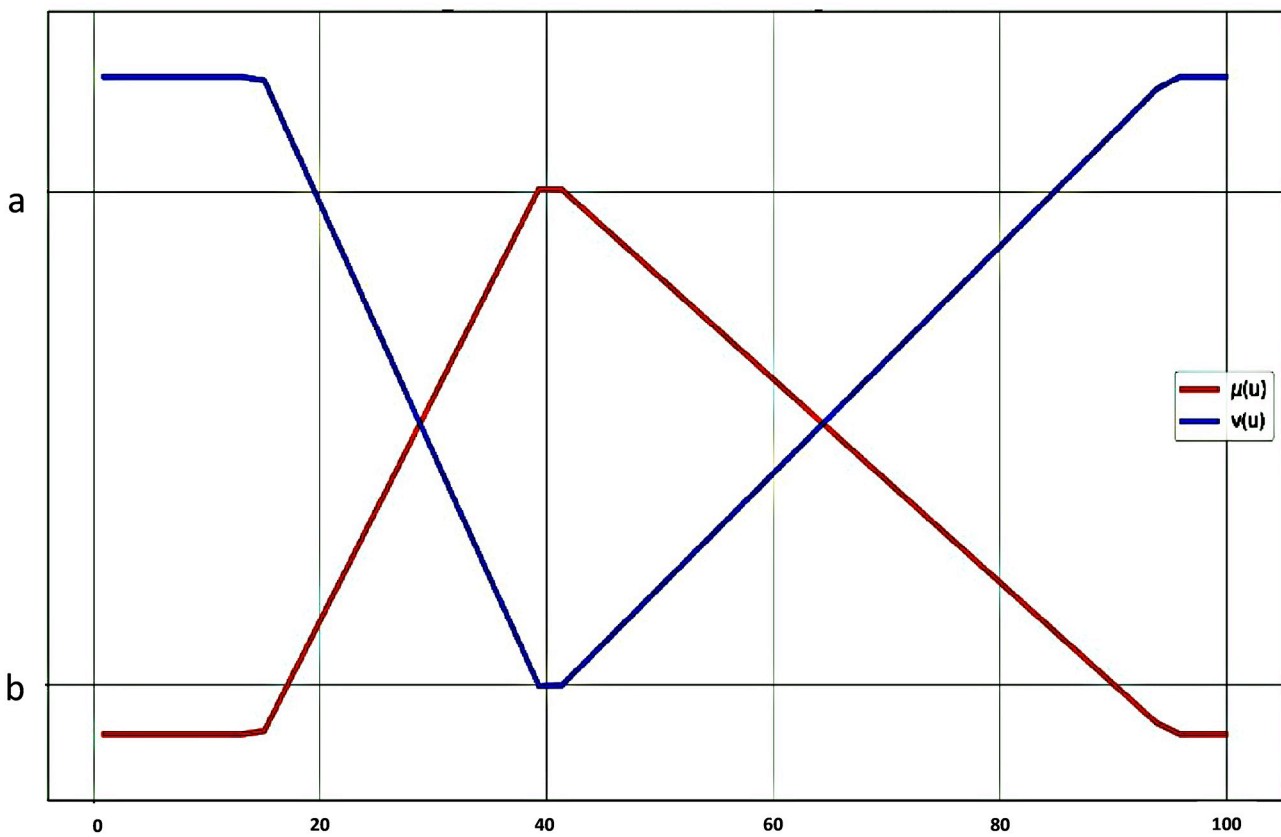

**Fig 3. TIFN membership and non-membership function.**

The intuitionistic fuzzy if-then rule is represented as follows [31]:

$$\text{If } x \text{ is } A(x) \text{ then } y \text{ is } B(y)$$

Thus, the intuitionistic fuzzy relation $R(x, y)$ to connect the input and output variables is given as follow:

$$R(x, y) = A(x) \rightarrow B(y)$$

The intuitionistic residual implication rule and the intuitionistic triple-I implication if-then rule is defined according to the following equations [31]:

$$R_R(x, y) = A(x) \rightarrow_{\mathcal{L}} B(y)$$

*2.3.5.4 Intuitionistic defuzzification.* Converting the fuzzy set to the crisp values is recognized as the last step for easier understanding and decision-making in the fuzzy control system. Therefore, it is necessary to use defuzzification methods. One of the most common defuzzification techniques is the center of gravity defuzzifier, which is used for the intuitionistic defuzzification step in this study. Thus, the center of gravity defuzzifier for *A* being an

**Table 2. Linguistic values and the parameters of membership and non-membership functions of TIFNs for two input variables: Severity and probability.**

| | Input variable | | TIFN | |
|---|---|---|---|---|
| | Probability | Severity | Membership (*a, b, c; α*) | Non-membership (*a, b, c; β*) |
| 1 | Rare | Insignificant | (0,0,0.5; 0.85) | (0,0,0.5; 0.1) |
| 2 | Unlikely | Minor | (0,1,1.5; 0.85) | (0,1,1.5; 0.1) |
| 3 | Possible | Moderate | (1,2.5, 3; 0.85) | (1,2.5, 3; 0.1) |
| 4 | Likely | Major | (2,3.5,4.5; 0.85) | (2,3.5,4.5; 0.1) |
| 5 | Almost Certain | Catastrophic | (4,5,5; 0.85) | (4,5,5; 0.1) |

intuitionistic fuzzy set on $X$ and $\lambda \in [0.1]$ is defined as [31]:

$$ICOG(A)^\lambda = \lambda \frac{\int X^{x \cdot \mu_A(x)dx}}{\int X^{\mu_A(x)dx}} + (1-\lambda)\frac{\int X^{x \cdot \nu_A(x)dx}}{\int X^{\nu_A(x)dx}}$$

Where, $\lambda$ is the balancing parameter that combines the output value of the intuitionistic Gravity Center. In more detail, $\lambda$ and $1 - \lambda$ are called the firing strength for the membership and non-membership values.

## 2.4 HAZOP technique based on the intuitionistic fuzzy sets

The risk levels for deviations were determined and prioritized based on the integration HAZOP technique and the intuitionistic fuzzy sets. In the first step, we defined the necessary variables for this study. According to the standard definition of risk, probability and severity are introduced as two input variables. In addition, each deviation's risk level is considered an output variable. Moreover, the linguistic values and each variable's physical domain were determined and shown in Table 2. This study determined five linguistic variables for both input variables as a 5×5 risk matrix.

Furthermore, the linguistic values and their membership and non-membership functions using TIFNs are defined for the risk level as the output linguistic variable (Table 3). It is worth

**Table 3. Linguistic values and the parameters of membership and non-membership functions of TIFNs for the output variable and risk level.**

| Rank | Linguistic variables | TIFN | |
|---|---|---|---|
| | | Non-membership functions | Membership functions |
| 1 | Extremely Low | (1,1,1.75; 0.1) | (1,1,1.75; 0.85) |
| 2 | Very Low | (1.5,2,2.75 0.1) | (1.5,2,2.75; 0.85) |
| 3 | Almost Low | (2.5,3,3.75; 0.1) | (2.5,3,3.75; 0.85) |
| 4 | More or Less Low | (3.5,4,4.75; 0.1) | (3.5,4,4.75; 0.85) |
| 5 | Low | (4.5,5,5.75; 0.1) | (4.5,5,5.75; 0.85) |
| 6 | A little Low | (5.5,6,7.25; 0.1) | (5.5,6,7.25; 0.85) |
| 8 | Extremely Medium | (6.75,8,8.75; 0.1) | (6.75,8,8.75; 0.85) |
| 9 | More or Less Medium | (8.5,9,9.75; 0.1) | (8.5,9,9.75; 0.85) |
| 10 | Vey Medium | (9.5,10,11.25; 0.1) | (9.5,10,11.25; 0.85) |
| 12 | Medium | (10.75,12,13.75; 0.1) | (10.75,12,13.75; 0.85) |
| 15 | More or Less High | (13.5,15,15.75; 0.1) | (13.5,15,15.75; 0.85) |
| 16 | High | (15.5,16,18.75; 0.1) | (15.5,16,18.75; 0.85) |
| 20 | Very High | (17.5,20,24.25; 0.1) | (17.5,20,24.25; 0.85) |
| 25 | Extremely High | (23.5,25,25; 0.1) | (23.5,25,25; 0.85) |

noting that instead of defining just four values for risk levels, as in the fuzzy approach, here we described 14 values for the risk level.

Then, the membership and the non-membership functions for each linguistic variable were defined using TIFN. Furthermore, Figs 4 and 5 show the membership and non-membership curve plots of our defined TIFNs for the input and output variables.

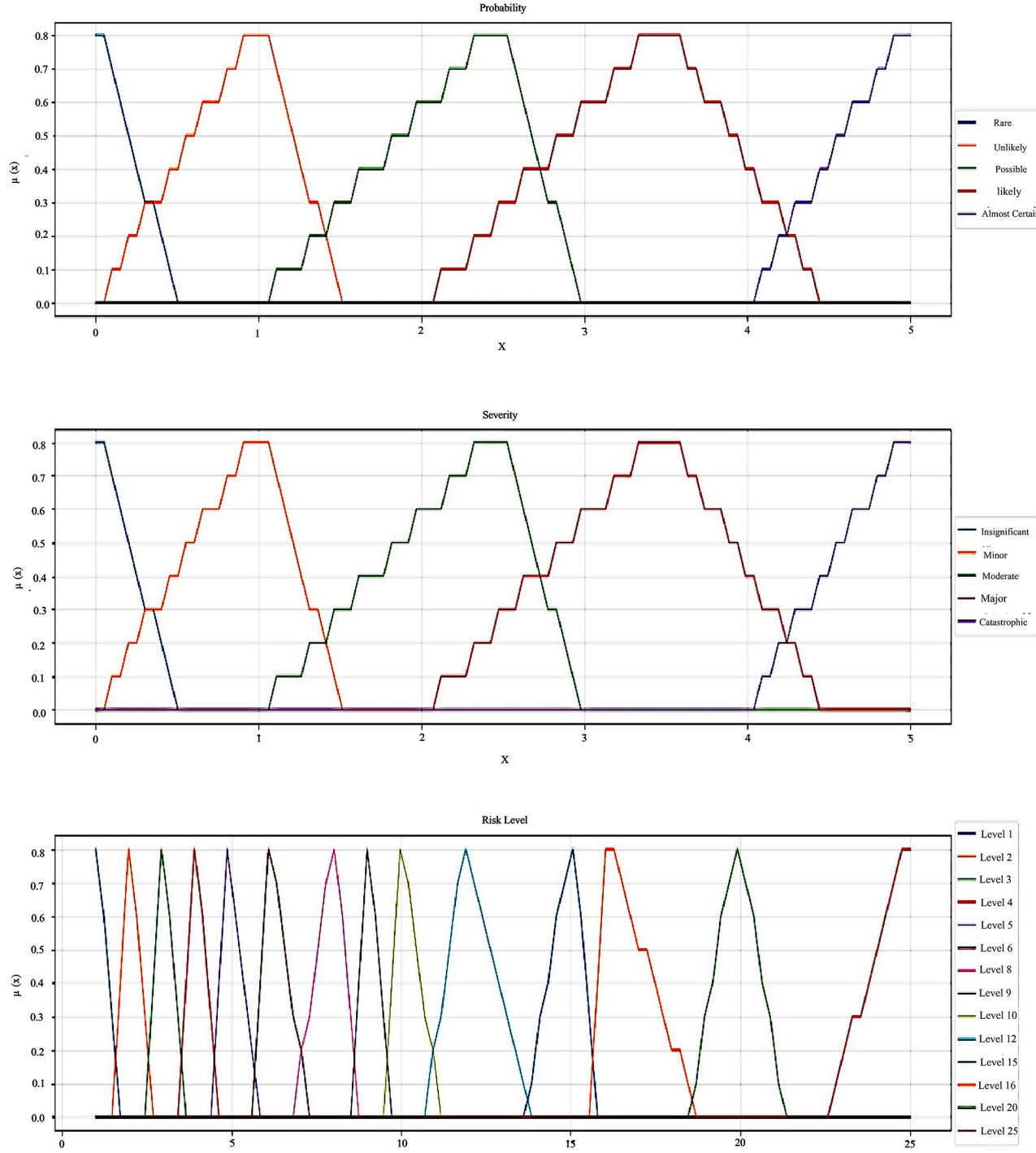

**Fig 4. The membership curve plots of TIFNs for two input variables: Severity and probability and the output variable: Risk level.**

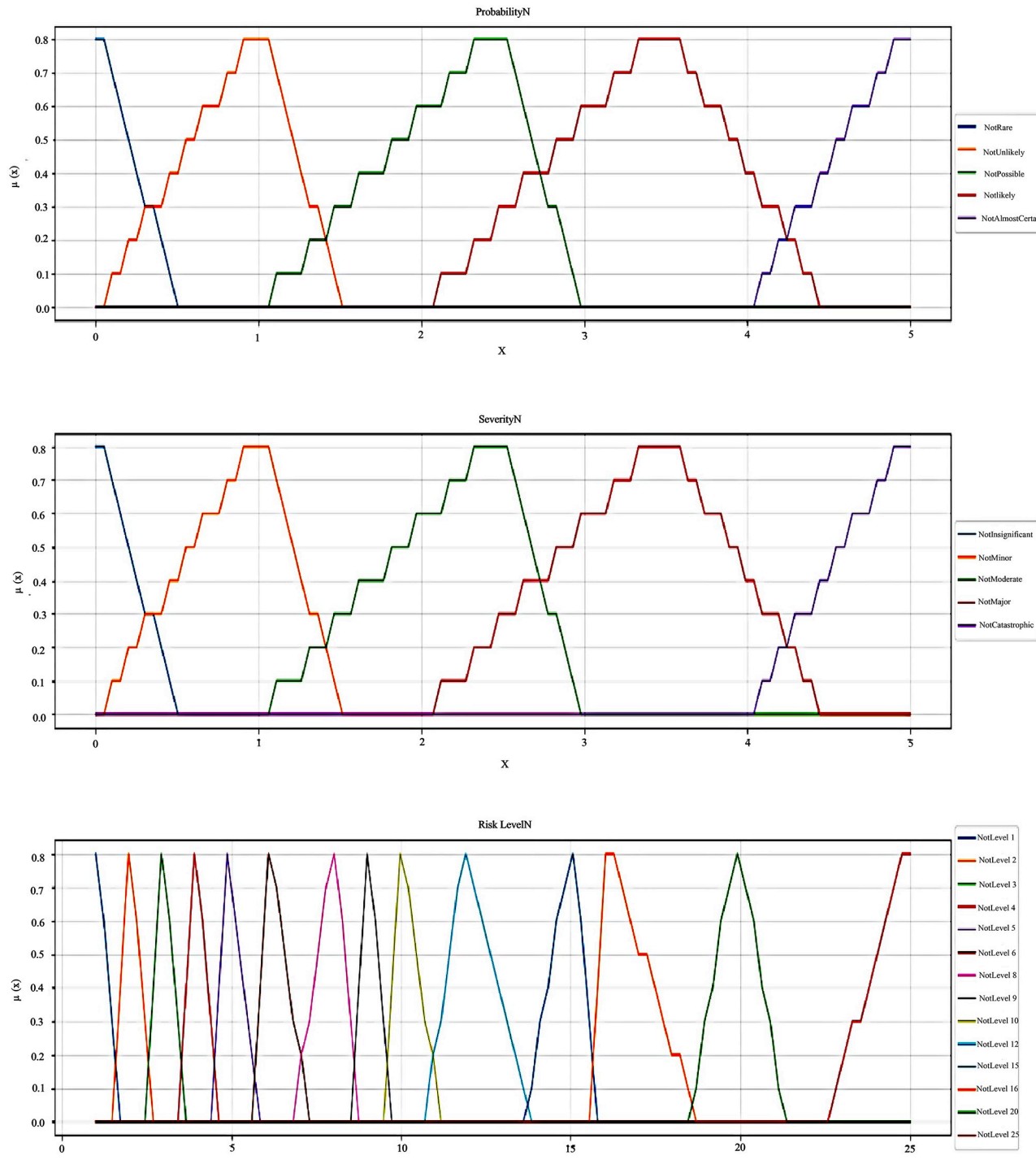

**Fig 5. The non-membership curve plots of TIFNs for two input variables: Severity and probability and the output variable: Risk level.**

The number of rules is determined by considering the input variables and their corresponding linguistic values. However, this study applied the intuitionistic fuzzy inference system, the linear combination of two conventional fuzzy inference systems, one for membership and another for non-membership functions. There are twenty-five rules for the membership

functions and twenty-five for the non-membership functions, which are written separately, equal to fifty rules. Each rule defines the relation between the probability and severity indices as input variables and the risk level as the output variable.

For instance:

If "probability" is "Very High 'and the "severity" is "Very High", then "Risk Level" is "High". Here the "Probability" and "Severity" of a deviation are "Very High", and then the "Risk Level" is "High." In this instance, "Probability and "Severity" are the linguistic input variables that are expressed by the linguistic value "Very High". In addition, the "Risk Level" is an output variable by the linguistic value "High".

In the intuitionistic fuzzy process, we considered fourteen levels for the risk evaluation due to the overlap and uncertainty in the boundaries of the risk levels, which is slightly different from the standard risk matrix (there are only four risk levels in the traditional risk matrix). Besides, the gravity center is utilized for intuitionistic defuzzification to obtain crisp outputs. In this study, the value of the balancing parameter between membership and non-membership functions in the linear combination of these two fuzzy inference systems, λ, was considered 0.99. After the defuzzification step, the crisp values helped to rank and prioritize the risk levels. It should be noted the risk levels are from 1 to 25 in the intuitionistic fuzzy approach, in which one is related to the lowest risk level, and 25 has the highest risk level in the risk matrix. Fig 6 shows the 3D surface plot of the output variable (Risk Level) versus the two input variables (Probability and Severity) obtained by our proposed intuitionistic fuzzy inference system. Fig 6 shows the 3D surface plot of the output variable (Risk Level) versus the two input variables (Probability and Severity) obtained from the intuitionistic fuzzy inference system.

## 3. Results

The PSA system under consideration in this study is classified into three nodes based on changes in features of operational parameters, including the temperature and pressure across the system (Fig 7). In this classification, there is the air compressor system at the first node, the dryer and Air receiver tank at the second node, and the oxygen generator and oxygen storage tank at the third node. It should be noted that there is compressed air along with the first and second nodes and condensed oxygen with a purity of over ninety percent in the third node.

The four process parameters in these three nodes include pressure (gas, oil), temperature (air, oil), flow, and voltage. The deviations in the PSA system were constructed using three keywords, including more, less, and none. In the risk assessment process, nine deviations were identified to 103 causes in the PSA system, some of which are summarized in Table 4. The number of deviations for four parameters, including the pressure, temperature, flow, and voltage, was 3, 2, 2, and 2, respectively. More detailed information about the process parameters, keywords, and deviation in each node is presented in Table 4. It is pertinent to mention that some deviations like non-voltage and non-flow are not considered in this study because their consequences are similar to those with low parameters. In addition, we also determined the over-flow only in the first node because the increase in the flow in the other two nodes is a function of the first node. It means that there is no other specific factor to increase the flow in the second and third nodes. Besides, two operational parameters, voltage and flow were not considered in the third node because it is unlikely to happen.

According to the results, it can see that the compressor device at the first node with the highest deviations (6), followed by the dryer at the second node (2), and then the oxygen equipment at the third node (1) have the highest number of deviations, respectively. Table 5 shows the differences in risk levels between conventional and fuzzy HAZOP methods. The results of the other two nodes and the average risk levels of nodes are also presented in Table 5.

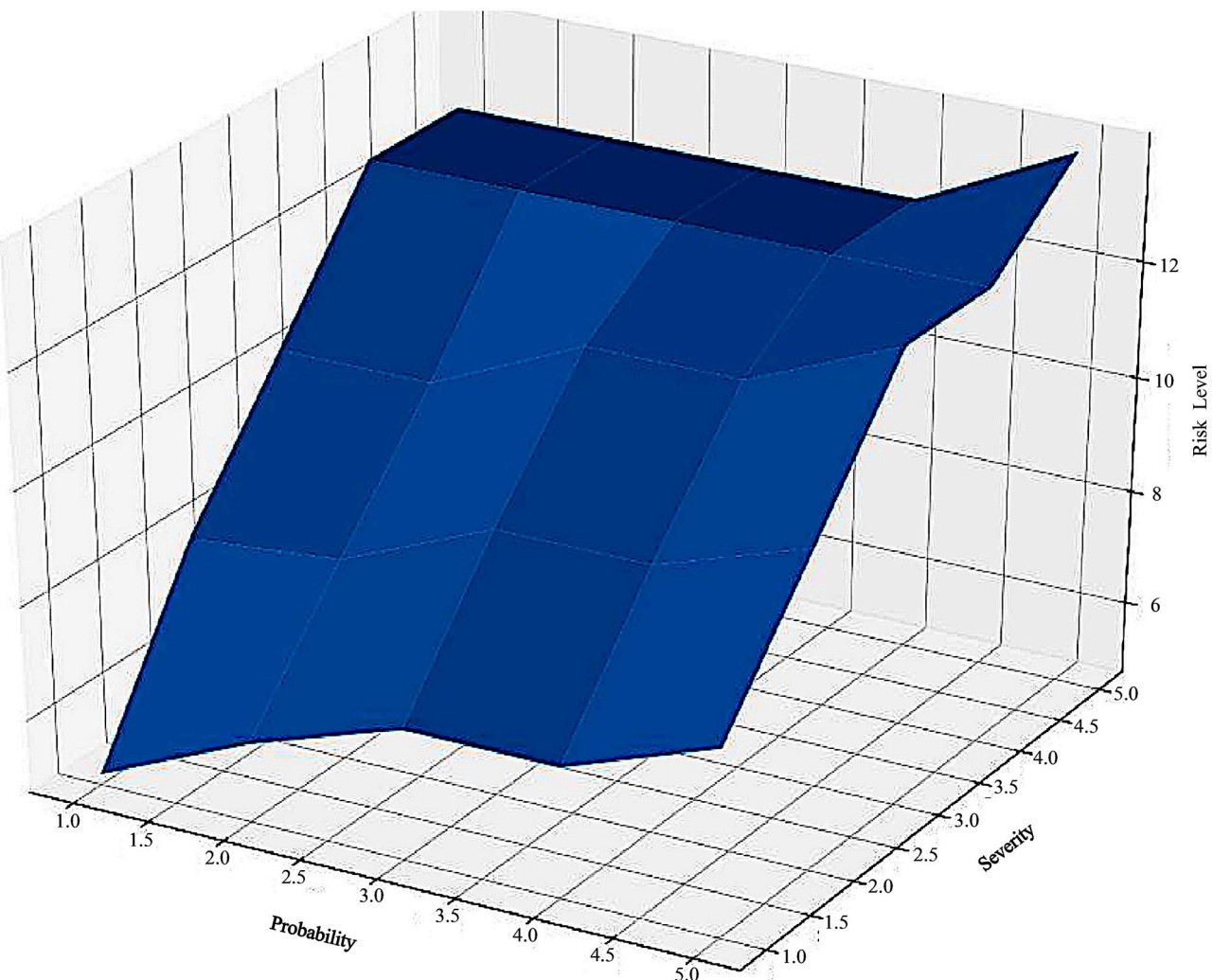

**Fig 6. The 3D surface plot of the output variable: Risk level, versus the two input variables: Probability and severity.**

The results also demonstrated that four out of 103 deviations are unacceptable risk levels, three related to the compressor unit. Finally, the comparison of results based on three nodes is shown in Fig 8.

The results of both approaches can be seen in Table 6. These findings are related to the fact that there are different impacts of probability and severity on the risk level. For instance, when the probability of the deviation is rare, and the severity is minor in the fuzzy approach, the risk value is equal to 5.39; but when the probability changes to the unlikely instead of rare; while the intensity becomes insignificant, the risk value increases from 5.39 to 7.45. While the risk values for these two situations are 2 in the conventional approach, where the first deviation is in the 2*1 level, and the second is in the 1*2 level. As shown in Table 6, the four most critical deviations are related to the increase in the process parameters, including pressure, temperature, and voltage, especially at the air compressor's first node, illustrated by the red color. In addition, it needs to be mentioned that all mentioned deviations in Table 6 (with over-guideword) could have resulted in the fire and explosion as final consequences on the system.

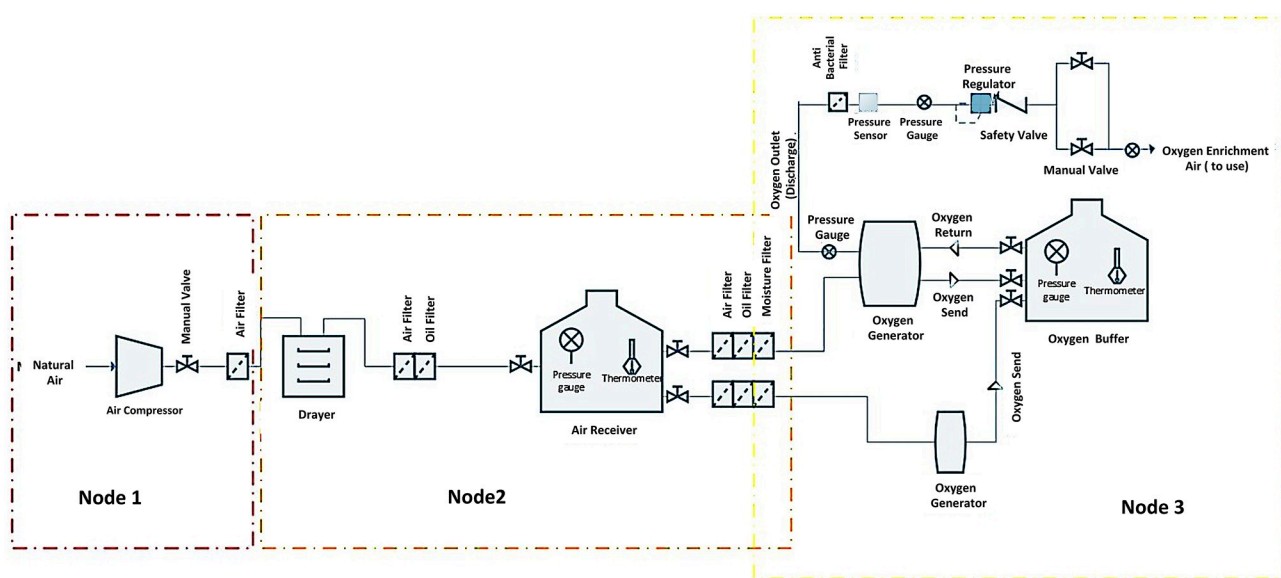

**Fig 7. Flowchart of hospital concentration system and node made based on HAZOP method.**

**Table 4. The information about the deviations in the PSA system (Based on the causes).**

|  | Process parameters | keywords | | | Nodes | | | deviations |
|---|---|---|---|---|---|---|---|---|
|  |  | More | Less | None | One | Two | Three | total |
| 1 | Pressure | 1 | 1 | 1 | 1 | 1 | 1 | 3 |
| 2 | Temperature | 1 | 1 | - | 1 | 1 | - | 2 |
| 3 | Flow | - | 1 | 1 | 2 | - | - | 2 |
| 4 | Voltage | 1 | 1 | - | 2 | - | - | 2 |

The relative frequency of the identified risks level in the conventional and intuitionist fuzzy approaches is presented in Fig 9, which compares the risk levels between the two adopted methods in this section. As shown in Fig 9, both conventional and intuitive fuzzy techniques have similar results at unacceptable risk levels. It means the risk values are equal in the fourth level in the traditional method and the first level in the intuitive fuzzy system. On the other hand, these results are different from other risk levels. For example, there is more than a six percent difference in the low-level risks between the fuzzy and conventional approaches (the first level in the traditional method and the fifth level in the intuitive fuzzy system). This

**Table 5. Frequency distribution of risks in conventional and intuitive fuzzy HAZOP approaches in three nodes.**

| Nodes | Approaches | First | Second | Third | Fourth | Fifth |
|---|---|---|---|---|---|---|
| Node1 | Conventional approach | 0 | 19 | 30 | **3** | - |
| Node1 | Intuitive fuzzy approach | **3** | 13 | 17 | 15 | 4 |
| Node2 | Conventional approach | 2 | 10 | 20 | 0 | - |
| Node2 | Intuitive fuzzy approach | 0 | 12 | 8 | 8 | 4 |
| Node3 | Conventional approach | 0 | 11 | 7 | **1** | - |
| Node3 | Intuitive fuzzy approach | **1** | 5 | 2 | 8 | 3 |

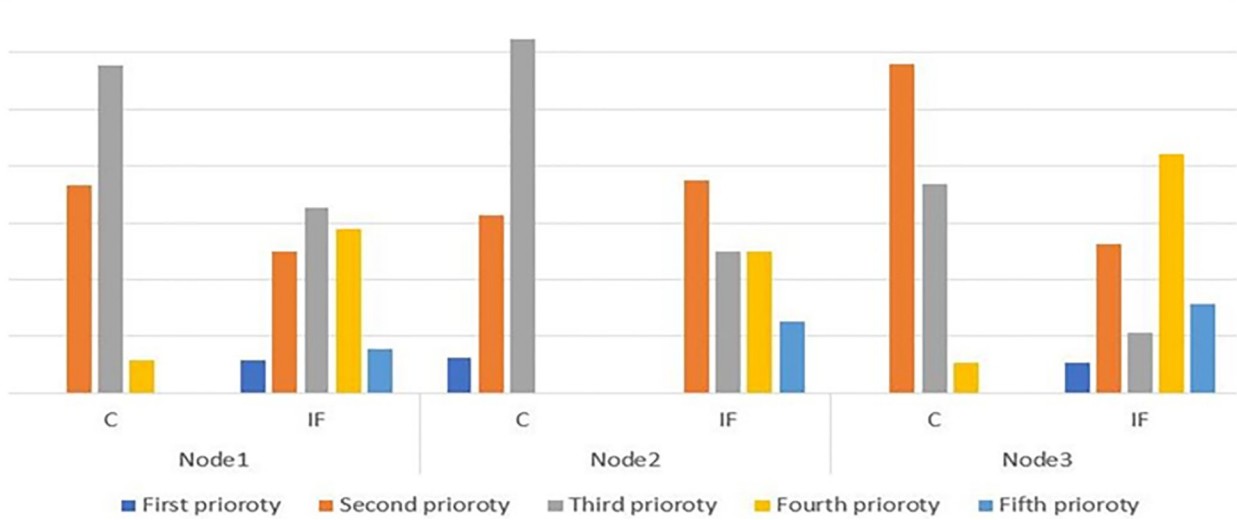

**Fig 8. Relative abundance distribution of risk level of deviations detected in conventional (C) and fuzzy approach (IF).**

difference can also be seen in the middle levels. As can be seen, most of the risk levels in the conventional approach fell into the third category as medium risk levels, contrary to the fuzzy method. It must be said the risk level categories in the fuzzy approach are defined in more levels than the conventional method, so the three middle levels have relatively closer numerical results to each other.

## 4. Discussion

This study demonstrated an attempt to utilize the HAZOP technique to determine the most potential deviations for the PSA system in a hospital. In this regard, Lee and Chang) 2014) [34] and Marhavilas et al.) 2020) [26] utilized the HAZOP approach to identify possible deviations for storage tanks containing flammable substances in petroleum industries. Similarly, Riad et al.) 2020) [35] used the HAZOP method to determine the potential hazards of the LPG spherical storage tanks. Moreover, the study conducted by Hu et al.) 2009) [36] showed that the HAZOP approach has sufficient validity for identifying the potential risks in a Gas Turbine Compressor. Thus, we attempted to identify the most critical deviations (9 deviations with 103 causes) in the medical oxygen production system through the HAZOP method with a preventive strategy.

One of the criticisms of the conventional methods in risk assessment is that data are subjective, unreliable, and static. Thus, dynamic processes have been developed for risk assessment [37]. In this regard, Cheraghi, Eslami Baladeh, and Khakzad) 2019) [38] integrated the HAZOP with the AHP & TOPSIS fuzzy approach to reduce the limitations of the conventional methods. The researchers concluded that although performing a dynamic process is more time-consuming, it can lead to better hazard identification and logical distribution of risk priority levels. Similarly, Marhavilas et al.) 2020) [26] applied the fuzzy-based HAZOP method to make decisions; it concluded that the fuzzy approach has more preference in the decision-making process and prioritization of hazard ranking than the traditional approach, especially under uncertain conditions. It is thus essential to have a dynamic approach to prioritizing the determined risks for more appropriate assessment. In addition, Jianxing et al.) 2019) [39] presented a fuzzy Fault Tree Analysis based on the weakest t-norm operators for assessing the

**Table 6. Four critical deviations in the conventional (C) and fuzzy (IF) approaches in the PSA system.**

| No | Node | Guideword | Parameter | Deviation | Cause | Effects | Control/Existing | Control/suggested | Risk | |
|----|------|-----------|-----------|-----------|-------|---------|------------------|-------------------|------|------|
| | | | | | | | | | C | IF |
| 1 | 1 | Over | Pressure | Over pressure of gas | Failure and burning of the solenoid valve | Not working the solenoid valve, not opening inlet valve, and raising internal pressure | Visual inspection-alarm of device | Replacement of parts according to the repair program | 12 | 9.95 |
| 2 | 1 | Over | Pressure | Over pressure of oil | Failure of the pressure switch in the fan of compressor | raising the oil temperature—lack of ventilation and air circulation in the condenser | Visual inspection-the alarm of device | Ensuring of connection the voltage to pressure switch—replacement if necessary | 9 | 8.95 |
| 3 | 3 | Over | Pressure | Over pressure of gas | Improper adjustment or defect of the pressure switch in Oxygen Generator | Raising the pressure of the oxygen more than the pressure point of the switch. not turning off the Oxygen Generator | Sensors and Thermometers, visual inspection | Replacement of parts according to the repair program | 15 | 13.94 |
| 4 | 3 | Over | Pressure | Over pressure of gas | Failure of the pneumatic valve | Defects in the air cycle inside the adsorbent column—reducing the oxygen purity | Check the purity of the output oxygen | Replacement of parts according to the repair program | 6 | 7.45 |
| 5 | 3 | Over | Pressure | Over pressure of gas | Malfunction of the outlet pressure regulator | Failure to reduce the oxygen pressure for consumption, the tension on the inner wall of the pipelines, and the possibility of oxygen leakage | visual inspection | Replacement of parts according to the repair program | 8 | 8.95 |
| 6 | 1 | Over | Voltage | Over voltage | Failure and burning of solenoid valve | Improper operation the switching on/off of current-raising internal temperature | Existing instructions—visual inspection—the alarm of the device | Use of voltage regulator inside the network—replacement of defective part | 15 | 13.94 |
| 7 | 1 | Over | Voltage | Over voltage | Failure in the starter relay in the compressor | Raising the temperature—having engine sound but not working | Existing instructions—visual inspection—the alarm of the device | Use of voltage regulator inside the network—replacement of defective part | 12 | 9.95 |
| 8 | 1 | Over | Voltage | Over voltage | Failure of the overload device due to the high heat of the engine | Shutdown and burnout of the engine | Existing instructions—visual inspection—the alarm of the device | Use of voltage regulator inside the network—replacement of defective part | 9 | 8.95 |
| 9 | 1 | Over | Voltage | Over voltage | Failure of the board relay of the fan | Failure to supply the voltage of fan and to burn the fan | Existing instructions—visual inspection—the alarm of the device | Use of voltage regulator inside the network—replacement of defective part | 10 | 9.95 |
| 10 | 1 | Over | Voltage | Over voltage | Short circuit in the contactor due to power fluctuation-overheating its coil | Lack of protection of the motor against overload and current—creating a momentary pulse in the circuit—turning the engine continuously on and off. | Existing instructions—visual inspection–the alarm of the device | Use of voltage regulator inside the network—replacement of defective part | 12 | 9.95 |
| 11 | 1 | Over | Voltage | Over voltage | Short electrical connection inside the capacitor due to the loss of the inner wet layer | Creating the overload and burning the capacitor–failure the compressor motor- Condenser fan turns on but the compressor not start | Existing instructions—visual inspection–the alarm of the device | Use of voltage regulator inside the network—replacement of defective part | 15 | 13.94 |

*(Continued)*

**Table 6.** (Continued)

| No | Node | Guideword | Parameter | Deviation | Cause | Effects | Control/Existing | Control/suggested | Risk C | Risk IF |
|---|---|---|---|---|---|---|---|---|---|---|
| 12 | 1 | Over | Voltage | Over voltage | Overload relay does not cut off the circuit and does not operate in time by increasing voltage and heat | Damaging to the engine due to overload and burning the motor | Existing instructions —visual inspection– the alarm of the device | Use of voltage regulator inside the network— replacement of defective part | 6 | 7.5 |
| 13 | 1 | Over | Temperature | Over-temperature of oil | Dirty refrigeration components | raising the temperature of compressor oil and damaging to air conditioner components | Existing instructions —Visual inspection–the alarm of the device | Ensuring of connection, the voltage to the fan motor—replace if necessary | 12 | 9.95 |
| 14 | 1 | Over | Temperature | Over-temperature of output air | Low level of refrigerant in the compressor | Raising the temperature of compressor oil-Fault in drain of condensation | Visual inspection | Fix leakage -Refrigerant gas injection | 15 | 13.94 |

leakage of substances in the submarine pipeline; and concluded that this new method is more practical and valid than the fuzzy approach. Furthermore, Viegas et al.) 2020) [22] utilized the intuitionist fuzzy-based HAZOP technique for multi-criteria decision-making for process safety. They concluded that the definition of non-membership functions and membership for each variable in the inference system led to more output accuracy than conventional and fuzzy approaches in an imprecise environment. Thus, this study used the intuitive fuzzy logic based on t-norm operators to rank hazards because the input data were uncertain and subjective in making the decision. Another study conducted by Marhavilas et al. (2022) [40] used the integration of the HAZOP technique, safety level colored maps, and Analytical-Hierarchy-Process for risk assessment in a process industry. The researchers concluded the proposed approach led to better decision-making in limited time and financial conditions for prioritizing risks with more reliable results.

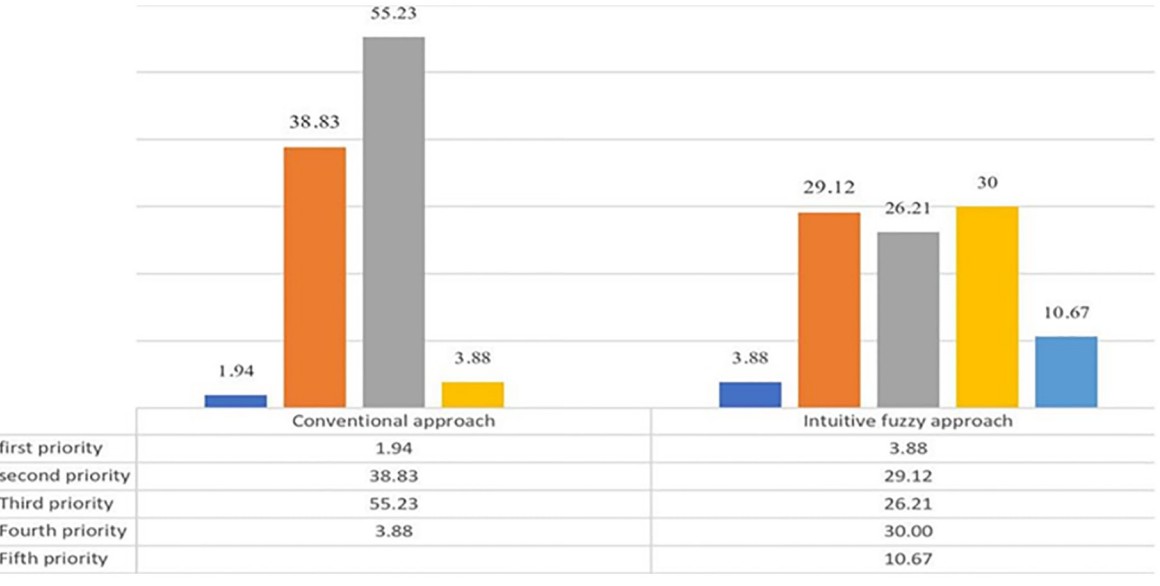

| | Conventional approach | Intuitive fuzzy approach |
|---|---|---|
| first priority | 1.94 | 3.88 |
| second priority | 38.83 | 29.12 |
| Third priority | 55.23 | 26.21 |
| Fourth priority | 3.88 | 30.00 |
| Fifth priority | | 10.67 |

**Fig 9. Relative frequency of risk level of deviations detected in convectional and fuzzy approach.**

In this study, two different levels were considered for prioritizing the risks. Firstly, the conventional HAZOP approach prioritized the risk levels by crisp numbers into four groups according to the standard risk matrix. Secondly, the intuitionistic fuzzy set's output was classified into five classes; there were no clear boundaries between the two adjacent levels, and they had a high overlap. As a result, the frequency and prioritization of the risk levels had a more rational logic and distribution, which distinguished from each other with more clarity. The method of this study was almost similar to the study by Ahn and Chang) 2016) [41], which used two different matrices in the conventional and fuzzy techniques for risk assessment in the chemical industry. The levels increased from three to seven classes in the fuzzy approach, leading to a more detailed rational prioritization of the risk levels.

In the present study, one of the primary deviations in the PSA system was related to the over-pressure in the air compressor and generators. In line with the finding of this study, Feiz Arefi et al.) 2020) [42] found that the most critical consequence of a central oxygen unit in the hospital systems is the probability of oxygen leakage due to an increase in pressure. According to the mentioned study, the central oxygen system in a hospital is evaluated by an integrated method, including Fault Tree Analysis and Layer of Protection Analysis; oxygen leakage due to overpressure is a critical risk that leads to the occurrence of fire and explosion.

Based on the findings, one of the most critical deviations in this study is over-pressure in the PSA system due to compressor failure, which may increase the likelihood of fire and explosion. In the meantime, the compressor capacitor is the main factor in managing engine torque and regulating the internal temperature and pressure in the compressor [43]. The solenoid valve's optimal performance depends on the compressor capacitor's correct operation. The solenoid valve is responsible for managing the air intake valve to the rotary part. On the other hand, during a solenoid valve malfunction, the internal valve does not work in proportion to the changes in pressure. It ultimately leads to an increase or decrease in internal pressure in the compressor.

Furthermore, the increase in internal pressure inside the absorber columns in the generator unit is associated with the defect in the pressure switch. The function of the pressure switch is to shut down the system while the internal pressure inside the concentrator reaches a certain level and then transfer the compressed air to another column. This instrument switches continuously between absorber columns. Therefore, its poor performance causes to increase in the over-pressure in the generator. As a result, it is necessary to control the pressure in the PSA system through specific instrumentations. Therefore, it is recommended to maintain the proper performance of solenoid valves and pressure switches to control the internal pressure across the PSA system, especially in compressors and generators.

Failures in the Solenoid valve can be resulted from an increase in the electrical resistance and internal temperature, leading to more heat and raising the average internal temperature in the system [44]. A study also conducted by Kim and Jeong (2014) [45] showed that the failure of the capacitor could lead to short circuits, overheating of the electric motor, and eventually burning off the compressor and even fire. This finding agrees with a study conducted by Rispoli et al. (2014) [46] which found that the leakage in the medical environment causes fires due to the electrical defects in the medical oxygen equipment. In addition, The study conducted by Eaton, Rama, and Macak (2006) [47] showed that capacitor failure and voltage changes could affect the refrigerant circulation system and oil circulation system in the compressor, which could lead to an increase in the internal temperature in the rotary part of the air compressor. Thus, the rising temperature is considered a critical deviation in the PSA system, which can harm the proper operation of the compressor. In general, increasing the PSA system's internal temperature due to causes such as internal defects or storage environment can be catastrophic. For instance, the proximity of oxygen production tank pipes to boilers,

improper installation, and improper maintenance of the oxygen supply system led to a fire in a hospital in South Korea in 2019 [48]. Also, the proximity of oxygen lines in the vicinity of hot water radiators and raising the internal temperature led to a fire in a hospital in India in 2019 [49]. It should be noted that the over-temperature and electronic equipment defects are directly related [45]. This finding is consistent with a study by Paul et al. (2020) [50] in India, who showed the PSA process produces heat that can damage electronic components and cause fires. Therefore, ventilation and cooling of compressors are essential aspects of this system.

Moreover, overvoltage was another significant deviation due to the defects of the electronic equipment in the PSA system. Electronic components are a vital part of air compressors among the equipment in the PSA system. Thus, the highest deviation of overvoltage is related to this unit. If the relay and other disconnecting parts do not operate in time or correctly, the input voltage to the electric motor will increase, and then the temperature will rise. According to the findings, the air compressor and its electronic equipment are recognized as crucial nodes in the PSA system. In the compressor, three critical deviations, including the over-pressure, over-temperature, and over-voltage, are directly related to the defect of the electronic board and its related components in the compressor that have an essential role in controlling the operating parameters in a safe range. In addition, it should be noted that all three of these parameters directly affect each other, and increasing one parameter increases other parameters.

This study tried to show the potential hazards in the oxygen production system, which can be generalized to other medical centers with similar equipment. Wood, Hailwood, and Koutelos (2021) [48] performed a study to reduce the risk of fires and explosions related to the oxygen system, especially during the corona epidemic and the increasing demand for high-purity oxygen for Covid-19 patients in hospitals. They have found that hospital fires may be prevented through risk assessment techniques and awareness of the potential hazards of oxygen. In this regard, a risk-based maintenance study for critical care medical equipment was conducted by Vala et al. (2018) [51]. This study showed the crucial role of the maintenance activities of hospital equipment in providing sustainable services to patients, which finally led to a reduction of maintenance-associated costs and better medical equipment life. Similarly, a study conducted by Corciová, Andritoi, and Ciorap (2013) [52] attempted to ensure the quality of medical equipment in the hospital environment. It has been found that risk assessment of hospital equipment has an essential role in preventing accidents and maintaining patient safety. Overall, the findings of these studies suggest that risk assessment can be used to control or eliminate crucial deficiencies and their underlying causes and deal with potential accidents by developing an effective safety program. As a result, managers and planners of the medical units can use the findings in this study to better prepare the hospital crisis management, which provides data about the critical deviations and the severity of the hospital oxygen equipment.

The study by Mostert and Coetzee (2014) [8] showed that the possibility of oxygen system failure and then oxygen leakage is unlikely, but it is potentially hazardous for medical centers. Therefore, it is necessary to consider a comprehensive and systematic program to detect and manage defects in this system by a proactive approach. The survey by Porte et al. (2018) [53] also showed a significant relationship between the risk assessment of medical equipment and staff awareness that can be used to hold training programs. Generally, the findings of this study can provide potential deviations as the minimum requirements that a hospital equipped with the PSA oxygen unit needs to know for the maintenance schedules and prevent malfunctions as possible. By considering other critical hospital equipment, such as anesthesia suction systems and boilers, as process equipment, this method can be successfully used to identify potential critical risks and events.

By considering the emergence of new methods, it is suggested that the results of this study be analyzed and compared using the third type of fuzzy logic, one of the new generalizations of the first type of fuzzy method. According to the studies [54, 55], the third type of fuzzy logic has more degrees of freedom and non-deterministic upper and lower limits for the uncertainty degree, which is suitable in uncertain situations to achieve the highest degree of flexibility. In addition, it is less sensitive to the changes in the parameters involved in a process; As a result, it leads to a more accurate prioritization of the risk deviations [54, 55]. One such example is the study by Mohammadzadeh et al. (2021) [55], which utilized the third type of fuzzy approach for a nonlinear dynamic system with uncertain parameters. Similarly, Taghieh et al. (2022) [56] investigated the voltage distribution problems in microgrids with this approach. Both of these studies implied that the third type of fuzzy approach could consider the impact of unknown external distributions for the dynamic assessment of micro-networks.

However, the study has some limitations that may have affected the findings. One of the limitations of this study was that the medical oxygen system under study is a young system with a lifespan of almost three years, which has led to a lack of data about the system's defects. We attempted to obtain data from other sources, such as articles and publications, and through a system with longer operating times to overcome this limitation. Another limitation of this study was the low experience and knowledge of maintenance operators about the oxygen system, which caused them not to have enough information about the system's potential hazards, technical errors, and dangerous failures. In addition, there was a lack of a written handbook for the medical oxygen system that included a written description of the information about the defects, problems, and repairs of the system. As a result, the operators presented all the information subjectively to the researchers. Furthermore, the HAZOP is a critical technique in determining the deviations in the system; but it also has one crucial drawback, including not determining the failure of the PSA equipment. Therefore, utilizing Failure Mode and Effect Analysis technique can be helpful.

## 5. Conclusion

Hospital fires are among the most critical concerns in medical centers, and medical equipment, especially medical oxygen concentrator units, is a potential fire source. Concerning the vital role of getting a proactive approach to fire management at the hospital, the present study aimed to assess the potential risk of fire-related medical oxygen production equipment using the PSA technology. The results showed the compressor has the most critical deviations among other equipment. The findings also showed that the intuitionistic fuzzy approach combined with the risk assessment is a proper tool in uncertain conditions, improving decision-making and leading to more precise and accurate results. The present study's findings can be a guideline for predictive maintenance optimization and planning requirement training programs for medical equipment management and maintenance staff.

## Acknowledgments

Hereby, we would like to express our most sincere gratitude to the manager of the studied hospital and the staff of the studied unit, who cooperated in the conduction of the study.

## Author Contributions

**Conceptualization:** Mahboubeh Es'haghi.

**Data curation:** Yeganeh Yousofnejad.

**Formal analysis:** Yeganeh Yousofnejad.

**Methodology:** Fatemeh Afsari.

**Software:** Fatemeh Afsari.

**Supervision:** Mahboubeh Es'haghi.

**Writing – original draft:** Yeganeh Yousofnejad, Mahboubeh Es'haghi.

**Writing – review & editing:** Mahboubeh Es'haghi.

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
