## [Decision Letter · Decision Letter 0]

11 Sep 2022

PONE-D-22-22776Dynamic risk assessment of hospital oxygen supply system by pressure swing absorbance technology based on the integration of HAZOP technique and intuitionistic fuzzy setPLOS ONE

Dear Dr. Esaghi,

Thank you for submitting your manuscript to PLOS ONE. After careful consideration, we feel that it has merit but does not fully meet PLOS ONE’s publication criteria as it currently stands. Therefore, we invite you to submit a revised version of the manuscript that addresses the points raised during the review process.

We look forward to receiving your revised manuscript.

Kind regards,

Ardashir Mohammadzadeh, Phd

Academic Editor

PLOS ONE

Journal Requirements:

When submitting your revision, we need you to address these additional requirements. 1. Please ensure that your manuscript meets PLOS ONE's style requirements, including those for file naming. The PLOS ONE style templates can be found at https://journals.plos.org/plosone/s/file?id=wjVg/PLOSOne_formatting_sample_main_body.pdf and https://journals.plos.org/plosone/s/file?id=ba62/PLOSOne_formatting_sample_title_authors_affiliations.pdf 2. During the internal evaluation of the study we noted that experts were included consisting of consist of occupational health and safety experts, artificial intelligence, medical equipment manager,  hospital crisis manager, and device operators to perform the risk assessment. As such please  provide additional details regarding participant consent. In the ethics statement in the Methods and online submission information, please ensure that you have specified whether consent was written or verbal/oral. If consent was verbal/oral, please specify: 1) whether the ethics committee approved the verbal/oral consent procedure, 2) why written consent could not be obtained, and 3) how verbal/oral consent was recorded. If your study included minors, please state whether you obtained consent from parents or guardians in these cases. If the need for consent was waived by the ethics committee, please include this information. 3. Thank you for stating the following in the Acknowledgments Section of your manuscript:  "This research was supported by Kerman University of Medical Sciences by Ethical Code (ID: IR.KMU.REC. 1399.500). " We note that you have provided funding information that is not currently declared in your Funding Statement. However, funding information should not appear in the Acknowledgments section or other areas of your manuscript. We will only publish funding information present in the Funding Statement section of the online submission form. Please remove any funding-related text from the manuscript and let us know how you would like to update your Funding Statement. Currently, your Funding Statement reads as follows:  "This research was supported by Kerman University of Medical Sciences by Ethical Code (ID: IR.KMU.REC. 1399.500).  " Please include your amended statements within your cover letter; we will change the online submission form on your behalf. 4. PLOS requires an ORCID iD for the corresponding author in Editorial Manager on papers submitted after December 6th, 2016. Please ensure that you have an ORCID iD and that it is validated in Editorial Manager. To do this, go to ‘Update my Information’ (in the upper left-hand corner of the main menu), and click on the Fetch/Validate link next to the ORCID field. This will take you to the ORCID site and allow you to create a new iD or authenticate a pre-existing iD in Editorial Manager. Please see the following video for instructions on linking an ORCID iD to your Editorial Manager account: https://www.youtube.com/watch?v=_xcclfuvtxQ 5. Please include your full ethics statement in the ‘Methods’ section of your manuscript file. In your statement, please include the full name of the IRB or ethics committee who approved or waived your study, as well as whether or not you obtained informed written or verbal consent. If consent was waived for your study, please include this information in your statement as well.

**Additional Editor Comments:**

-Compare you results with new related methods;

-Check the paper for writing errors;

-Add a direction for readers; Add some references and statements about the potential improvement by type-3 fuzzy logic systems;

-Add some diagrams for better understanding your method

Reviewers' comments:

Reviewer's Responses to Questions

**Comments to the Author**

1. Is the manuscript technically sound, and do the data support the conclusions?

Reviewer #1: Yes

Reviewer #2: Yes

2. Has the statistical analysis been performed appropriately and rigorously? 

Reviewer #1: N/A

Reviewer #2: Yes

3. Have the authors made all data underlying the findings in their manuscript fully available?

Reviewer #1: Yes

Reviewer #2: Yes

4. Is the manuscript presented in an intelligible fashion and written in standard English?

Reviewer #1: Yes

Reviewer #2: Yes

5. Review Comments to the Author

Reviewer #1: 1. the topic is not unique, but worthy of researching

2. there are hundreds of papers in google scholar about this topic since 2018

3. the title is too long. Please shorten it to be attractive and citable

4. the abstract is too long. Please shorten it and revise the plagiarism

5. the aim is clear

6. the KEYWORDS are good.

7. lack of the abbreviations section

8. the introduction provide sufficient background information for readers in the immediate field to understand the problem/hypotheses

9. the text arrangement is good

10. the logic is clear

11. the paper is not novel

12. there are few grammatical errors in this article

13. the related concepts are introduced

14. the readability is sufficient

15. the results are good

16. all figures/tables are clear enough to summarize the results for presentation to the readers

17. all figures/tables are well referred to in the text

18. the theoretical analysis in this article is sufficient

19. the discussion of results from multiple angles is sufficient

20. the conclusion are good

21. the reference section are good

22. the references are in order within the text

23. Bias is acceptable

24. Fund is mentioned

25. Ethical approval is not needed

26. Conflict of interest is mentioned

27. Acknowledgement is mentioned

28. You can use my suggestions

My final decision is acceptable after minor revision

Reviewer #2: Review on research titled” Dynamic risk assessment of hospital oxygen supply system…”

General comments:

Title: your title “Dynamic risk assessment of hospital oxygen supply system by pressure swing absorbance technology based on the integration of HAZOP technique and intuitionistic fuzzy set” is too long. Make it short and precise, maximum of 10-15 words.

I suggest “Dynamic risk assessment of hospital oxygen supply system” or “Dynamic risk assessment of hospital oxygen supply system by pressure swing absorbance technology” and Add where and when the research was done.

Abstract:

Background: The background of information of your abstract is too extended. Try to minimize it.

Methods: where and when you are assessing; is your research is qualitative or quantitative? Incorporate such information.

Result: what are the most frequent deviations you identified by HAZOP tool?

Introduction

Good introduction but your paragraphs (e.g. 1st paragraph) are too bulky and different issues are included in a single paragraph. So make your paragraph specific.

Result:

1. write all 9 deviation you found and report the with high magnitude as table 4

2. Your line 376 saying” The results showed some differences between the findings of both approaches, which can be seen in…. remove such expressions and just put (Table 6).

Suggestion: try to incorporate the above comments

Decision: Accepted with minor modifications

Reviewer: Mr Asamirew Alemu (BSc, MSc), Mizan-Tepi University, Ethiopia.

6. PLOS authors have the option to publish the peer review history of their article (what does this mean?). If published, this will include your full peer review and any attached files.

Reviewer #1: **Yes: **Hazim abdul rahman Alhiti

Reviewer #2: No

---

## [Author Response · Author response to Decision Letter 0]

19 Dec 2022

Responses to the Comments

Our responses are given in a point-by-point manner below. Changes to the manuscript are shown in the green color.

Editor Comments

A: We reviewed the manuscript according o the PLOS ONE's style requirements.

2. During the internal evaluation of the study we noted that experts were included consisting of consist of occupational health and safety experts, artificial intelligence, medical equipment manager, hospital crisis manager, and device operators to perform the risk assessment. As such please provide additional details regarding participant consent. In the ethics statement in the Methods and online submission information, please ensure that you have specified whether consent was written or verbal/oral. If consent was verbal/oral, please specify: 1) whether the ethics committee approved the verbal/oral consent procedure, 2) why written consent could not be obtained, and 3) how verbal/oral consent was recorded. If your study included minors, please state whether you obtained consent from parents or guardians in these cases. If the need for consent was waived by the ethics committee, please include this information.

A: In this study, the risk assessment of the oxygen generator was done. We did not have any human data or confidential information. Also, the consent of the participants in the meetings was verbal. 

 "This research was supported by Kerman University of Medical Sciences by Ethical Code (ID: IR.KMU.REC. 1399.500). "

 "This research was supported by Kerman University of Medical Sciences by Ethical Code (ID: IR.KMU.REC. 1399.500). "

A: We removed the funding-related text from the manuscript and added it in the title page. 

4. PLOS requires an ORCID ID for the corresponding author in Editorial Manager on papers submitted after December 6the, 2016. Please ensure that you have an ORCID iD and that it is validated in Editorial Manager. To do this, go to ‘Update my Information’ (in the upper left-hand corner of the main menu), and click on the Fetch/Validate link next to the ORCID field. This will take you to the ORCID site and allow you to create a new iD or authenticate a pre-existing iD in Editorial Manager. Please see the following video for instructions on linking an ORCID iD to your Editorial Manager account: https://www.youtube.com/watch?v=_xcclfuvtxQ

A: We performed according to the comment for the ORCID iD.

A: We added the following sentence in the "method" section:

In this research, we assessed the risk assessment of an oxygen supply system with the PSA technology, which did not have human data or confidential information. Thus, the consent of the participants in the meetings was verbal. This research is supported by the Student Research Committee, Kerman University of Medical Sciences, by the Ethical Code (ID: IR.KMU.REC. 1399.500).

A: We reviewed and checked the reference list. 

Additional Editor Comments:

Compare you results with new related methods;

A: We added a new study in order to compare the method of this research with new related methods as following: 

Another study conducted by Marhavilas et al. (2022) (40) used the integration of the HAZOP technique, safety level colored maps, and Analytical-Hierarchy-Process for risk assessment in a process industry. The researchers concluded the proposed approach led to better decision-making in limited time and financial conditions for prioritizing risks with more reliable results.

Check the paper for writing errors

A: We checked the manuscript and reviewed by a translation academy. 

Add a direction for readers; Add some references and statements about the potential improvement by type-3 fuzzy logic systems;

A: We used two sources about the Type-3 fuzzy sets in the manuscript. In this regard, we suggested using this technique for future studies as following:

By considering the emergence of new methods, it is suggested that the results of this study be analyzed and compared using the third type of fuzzy logic, one of the new generalizations of the first type of fuzzy method. According to the studies (55, 54), the third type of fuzzy logic has more degrees of freedom and non-deterministic upper and lower limits for the uncertainty degree, which is suitable in uncertain situations to achieve the highest degree of flexibility. In addition, it is less sensitive to the changes in the parameters involved in a process; As a result, it leads to a more accurate prioritization of the risk deviations (55, 54). One such example is the study by Mohammadzadeh et al. (2021) (55), which utilized the third type of fuzzy approach for a nonlinear dynamic system with uncertain parameters. Similarly, Taghieh et al. (2022) (56) investigated the voltage distribution problems in microgrids with this approach. Both of these studies implied that the third type of fuzzy approach could consider the impact of unknown external distributions for the dynamic assessment of micro-networks.

Add some diagrams for better understanding your method.

A: We added one diagram about the total numbers of deviations in each node in the manuscript.

Finally, the comparison of result based on three nodes is shown in figure 8.

Reviewer #1

3. The title of the manuscript has been shortened according to the following: 1. The topic is not unique, but worthy of researching

2. There are hundreds of papers in Google scholar bout this topic since 2018

3. The title is too long. Please shorten it to be attractive and citable

4. The abstract is too long. Please shorten it and revise the plagiarism.

A: According to the comment, the abstract has shorted.

Events such as oxygen leakage in the oxygen generation systems can have severe consequences, such as fire and explosion. In addition, the disruption in the oxygenation systems can lead to a threat to patients' lives. Thus, this study aimed to identify the significant deviations in the oxygen supply system as critical equipment at hospitals based on the Hazard and Operability (HAZOP) method. Despite the advantages of risk assessment techniques, hazard identification techniques are still being utilized with deterministic and unreliable values and have a completely static nature. Therefore, using dynamic techniques to overcome intrinsic ambiguity in the risk assessment process through fuzzy sets has been recommended. Additionally, we proposed the HAZOP methodology to integrate with the intuitionistic fuzzy system for assessing the medical oxygen supply system using Pressure Swing Absorbance technology as a proactive approach. The results showed that the intuitionistic fuzzy approach, combined with the risk assessment method, is a suitable tool to eliminate uncertainty, improve decision-making, and result in more detailed and accurate findings. The approach adopted in this study can be used as a needs assessment tool to optimize maintenance programs and provide the necessary training for the staff, maintenance operators, and medical equipment managers.

7.

A: According to the comment, the abbreviations added in the manuscript.

Abbreviations: HAZOP, Hazard and Operability Study; PSA, pressure swing absorbance; IFS, intuitionistic fuzzy set. 5. The aim is clear

6. The KEYWORDS are good.

7. Lack of the abbreviations section

8. the introduction provide sufficient background information for readers in the immediate field to understand the problem/hypotheses

A: According to the comment, the manuscript was reviewed by a translation academy. 9. The text arrangement is good

10. The logic is clear

11. The paper is not novel

12. There are few grammatical errors in this article

13. The related concepts are introduced

14. The readability is sufficient

15. The results are good

 16. All figures/tables are clear enough to summarize the results for presentation to the readers

17. All figures/tables are well referred to in the text

18. The theoretical analysis in this article is sufficient

19. The discussion of results from multiple angles is sufficient

20. the conclusion are good

21. The reference section is good

22. The references are in order within the text

23. Bias is acceptable

24. Fund is mentioned

25. Ethical approval is not needed

26. Conflict of interest is mentioned

27. Acknowledgement is mentioned

28. You can use my suggestions

Reviewer #2:

Title: your title “Dynamic risk assessment of hospital oxygen supply system by pressure swing absorbance technology based on the integration of HAZOP technique and intuitionistic fuzzy set” is too long. Make it short and precise, maximum of 10-15 words.

I suggest “Dynamic risk assessment of hospital oxygen supply system” or “Dynamic risk assessment of hospital oxygen supply system by pressure swing absorbance technology” and Add where and when the research was done.

A: According to the comment, the second proposed title has selected

Dynamic risk assessment of hospital oxygen supply system by pressure swing absorbance technology.

Background: The background of information of your abstract is too extended. Try to minimize it.

A: According to the comment, the second proposed title has selected

Events such as oxygen leakage in the oxygen generation systems can have severe consequences, such as fire and explosion. In addition, the disruption in the oxygenation systems can lead to a threat to patients' lives. Thus, this study aimed to identify the significant deviations in the oxygen supply system as critical equipment at hospitals based on the Hazard and Operability (HAZOP) method. Despite the advantages of risk assessment techniques, hazard identification techniques are still being utilized with deterministic and unreliable values and have a completely static nature. Therefore, using dynamic techniques to overcome intrinsic ambiguity in the risk assessment process through fuzzy sets has been recommended. Additionally, we proposed the HAZOP methodology to integrate with the intuitionistic fuzzy system for assessing the medical oxygen supply system using Pressure Swing Absorbance technology as a proactive approach. The results showed that the intuitionistic fuzzy approach, combined with the risk assessment method, is a suitable tool to eliminate uncertainty, improve decision-making, and result in more detailed and accurate findings. The approach adopted in this study can be used as a needs assessment tool to optimize maintenance programs and provide the necessary training for the staff, maintenance operators, and medical equipment managers.

Introduction: Good introduction but your paragraphs (e.g. 1st paragraph) are too bulky and different issues are included in a single paragraph. So make your paragraph specific.

A: According to the comment, the paragraphs of introduction section are separated.

Methods: where and when you are assessing; is your research is qualitative or quantitative? Incorporate such information.

As a qualitative approach, the HAZOP is a systematic and structured technique that evaluates and identifies potential process hazards that may present risks to people, equipment, environment, and operation.

Result

1. Write all 9 deviations you found and report the with high magnitude as table 4

in this study, we found …..

A: We added the following diagram in the manuscript. 

Finally, the comparison of result based on three nodes is shown in figure 8.

Result

2. Your line 376 saying” The results showed some differences between the findings of both approaches, which can be seen in…. remove such expressions and just put (Table 6).

A: According to the comment, this sentence is deleted and changed according to the following:

We presented these results of both approaches in Table 6.

Result

What are the most frequent deviations you identified by HAZOP tool?

A: According to the Table 6, we mentioned the most important deviations in the PSA system.

---

## [Decision Letter · Decision Letter 1]

12 Jan 2023

Dynamic risk assessment of hospital oxygen supply system by HAZOP and intuitionistic fuzzy

PONE-D-22-22776R1

Dear Dr. Esaghi,

We’re pleased to inform you that your manuscript has been judged scientifically suitable for publication and will be formally accepted for publication once it meets all outstanding technical requirements.

Kind regards,

Ardashir Mohammadzadeh, Phd

Academic Editor

PLOS ONE

Additional Editor Comments (optional):

Reviewers' comments:

Reviewer's Responses to Questions

**Comments to the Author**

1. If the authors have adequately addressed your comments raised in a previous round of review and you feel that this manuscript is now acceptable for publication, you may indicate that here to bypass the “Comments to the Author” section, enter your conflict of interest statement in the “Confidential to Editor” section, and submit your "Accept" recommendation.

Reviewer #1: All comments have been addressed

2. Is the manuscript technically sound, and do the data support the conclusions?

Reviewer #1: Yes

3. Has the statistical analysis been performed appropriately and rigorously? 

Reviewer #1: Yes

4. Have the authors made all data underlying the findings in their manuscript fully available?

Reviewer #1: Yes

5. Is the manuscript presented in an intelligible fashion and written in standard English?

Reviewer #1: Yes

6. Review Comments to the Author

Reviewer #1: thank you for revising the paper. I accept it now as it is. Congratulations. i attached my review file

7. PLOS authors have the option to publish the peer review history of their article (what does this mean?). If published, this will include your full peer review and any attached files.

Reviewer #1: **Yes: **Hazim abdul rahman Alhiti

---

## [Editor Report · Acceptance letter]

8 Feb 2023

PONE-D-22-22776R1 

Dynamic risk assessment of hospital oxygen supply system by HAZOP and intuitionistic fuzzy 

Dear Dr. Es'haghi:

I'm pleased to inform you that your manuscript has been deemed suitable for publication in PLOS ONE. Congratulations! Your manuscript is now with our production department. 

Kind regards, 

on behalf of

Dr. Ardashir Mohammadzadeh 

Academic Editor

PLOS ONE